# Open acid dissolution—Ammonia solution extraction—ICP OES rapid determination of 7 trace metal elements in soil

**Jiahan Wang**, Junqiao Long, Feng Yang, Xiujin Yang, Wenguang Jiao*, Cheng Huang*

Haikou Marine Geological Survey Center of China Geological Survey, Haikou, 571127, China

* 450733437@qq.com (WJ); hc_learn@126.com (CH)

## Abstract

To overcome the corrosion of hydrofluoric acid on the ICP OES injection system in the acid dissolution system, this paper makes some improvements based on the traditional open digestion. The improved method does not require the complete removal of hydrofluoric acid. After appropriate digestion of the sample with a mixed acid, the solution can be transferred to a colorimetric tube containing ammonium hydroxide solution to give the final volume for analysis. In this paper, two-point standard curves are plotted using soil standards and process blanks, which is not only convenient but also overcomes the interference of the matrix effect. Through continuous experiments, the preferred ratio of mixed acid is 3 mL nitric acid + 5 mL hydrofluoric acid, and the concentration of ammonia solution is 0.5%. The spectral lines of the measured elements V (292.4), Cr (283.5), Co (228.6), Ni (231.6), Cu (324.7), Zn (213.8) and Pb (220.3) were determined. The method quantification limits of the seven measured elements V, Cr, Co, Ni, Cu, Zn and Pb were 0.909, 4.32, 0.269, 0.261, 0.968, 3.69 and 2.64 $\mu g\ g^{-1}$, respectively, and the precision was 3.5%, 5.2%, 4.8%, 2.4%, 6.1% and 4.5%, respectively. After processing six national standard materials according to the experimental method, the measured values of each measured element were basically in agreement with the certified values, indicating that this method is fully feasible for the measurement of V, Cr, Co, Ni, Cu, Zn and Pb in soil. This method greatly improves the efficiency of pretreatment and is particularly suitable for analysing large batches of samples.

## Introduction

Soil is one of the most important material foundations for human survival, and protecting and restoring the soil environment is an important task for environmental protection. Among the inorganic pollutants in soil, trace metal contamination is particularly prominent [1–3]. Trace metals such as V, Cr, Co, Ni, Cu, Zn and Pb not only reduce soil fertility and crop yields, but also affect human health through the food chain [4–6]. Under certain conditions, trace metals stored in the soil can be released and washed into water bodies or be carried into the air with soil particles blown by the wind, causing secondary pollution [7–9].

**Data Availability Statement:** All relevant data are within the manuscript and its Supporting Information files.

**Funding:** This study was supported by the Geological Survey Project of China Geological

Survey (Project No. DD20220992). Cheng Huang, the project manager, was mainly responsible for the theoretical guidance and supervision of this study.

**Competing interests:** The authors have declared that no competing interests exist.

The detection of trace metals in soil is essential for the protection and remediation of the soil environment. Commonly used methods for detecting trace metals in soil include X-ray fluorescence (XRF) [10,11], inductively coupled plasma optical emission spectroscopy (ICP OES) [12–14], inductively coupled plasma mass spectrometry (ICP-MS) [15–17] and atomic absorption spectroscopy (AAS) [18,19]. Among these methods, ICP OES is widely used in soil trace metal analysis because of its ability to analyse multiple elements simultaneously, its fast detection speed and its low instrument cost.

There are several common pretreatment methods for measuring metal elements in soil, including high pressure closed acid dissolution [20,21], open acid dissolution [22,23], microwave digestion [24–26] and alkaline fusion [27,28]. Of these, the alkaline fusion method is less commonly used due to its high salt content and complex operation. High-pressure closed digestion and microwave digestion are not suitable for large-scale pretreatment due to the limited digestion tank volume and complex operation. Open acid digestion is currently the most widely used method, and the acid added to open acid digestion usually includes hydrofluoric acid, which is used to destroy the soil lattice. To avoid damage to the ICP OES injection system from hydrofluoric acid, the hydrofluoric acid must be completely removed during the pretreatment process before the acid solution is added for dissolution and subsequent measurement. The hydrofluoric acid removal process is time consuming and the dissolution process is also time consuming, resulting in low pretreatment efficiency. In addition, the multi-step process can introduce significant error.

This article proposes an improvement to the traditional open acid dissolution method for the determination of metal elements in soil. Instead of completely removing hydrofluoric acid during digestion, ammonia solution is added at the appropriate time during digestion and the solution is transferred to a volumetric flask for measurement. The added ammonia can neutralise the hydrofluoric acid in the digestion solution, greatly reducing the corrosion of the ICP OES injection system by fluoride ions under alkaline conditions [29]. This new approach improves efficiency and is particularly suitable for large sample analysis.

## Materials and methods

### Main instruments

ICAP-6300 ICP OES (Thermo Fisher Scientific), electric heating plate (LabTech).

The instrument has been optimised for the best working conditions as shown in Table 1.

### Main reagents and materials

The national standard substances for soil used in this study, including GSS1a, GSS2a, GSS3a, GSS4a, GSS5a, GSS8a and GSS2, were developed by the Institute of Geophysical and Geochemical Exploration of the Chinese Academy of Geological Sciences. These materials consist mainly of constituents such as $SiO_2$ (ranging from 56.6% to 72.9%), $Al_2O_3$ (ranging from 11.8% to 27.4%) and $Fe_2O_3$ (ranging from 2.63% to 18.1%), among others.

**Table 1. Working parameters of ICP OES.**

| Parameter | Value | Parameter | Value |
|---|---|---|---|
| Power/W | 1350 | Auxiliary gas flow/(L min$^{-1}$) | 0.5 |
| Carrier gas pressure/MPa | 0.24 | Number of repeated observations | 3 |
| Vertical observation height/mm | 13 | Integral time/s | 15 |
| Peristaltic pump speed/(r min$^{-1}$) | 50 | Wash time/s | 40 |
| Nebulizer flow/(L min$^{-1}$) | 0.7 | Stability time/s | 5 |

The ammonia solution used was of guaranteed reagent, and the ultrapure water had a resistivity of not less than 18 MΩ·cm.

## Experimental method

0.1000 g of soil sample was taken into a PTFE crucible and a small amount of water was added to wet the sample. Then 3 mL of nitric acid and 5 mL of hydrofluoric acid were added sequentially and the mixture was heated to 180˚C until about 2 mL of acid remained. The electric heating plate was then switched off and the solution was transferred to a volumetric flask and made up to 25 mL with 0.5% ammonia solution. The supernatant was measured after filtration or allowing to settle for a period of time. A blank solution was prepared simultaneously with the sample.

The national standard substance GSS2, digested simultaneously with the sample, was used as the peak and the blank solution as the valley. The two-point standard working curve for each element was obtained by measuring the solutions under the selected instrument operating conditions. The test solution was then measured under the same conditions.

## Results and discussion

### Calibration curves

Some researchers use standard materials instead of standard solutions to draw calibration curves [30], which has the following advantages over the latter: (1) the results can be read directly without data conversion; (2) it avoids the complicated preparation and measurement steps of series standard solutions; (3) it greatly reduces the interference caused by matrix effects; (4) it is convenient to redraw the calibration curves when the instrument signal drifts. In addition, one of the biggest advantages of ICP OES is its large linear range, which means that calibration curves can be drawn with fewer concentration points. Therefore, in this study, the calibration curves were drawn using the national standard substance GSS2 as the high point and the blank solution as the low point.

### Concentration of ammonia solution

Four different concentrations of ammonia solution (0.1%, 0.5%, 1% and 2%) were tested for their re-solubilising effect on the national standard substance GSS1a, and the absolute values of the relative errors between the measured values of each tested element and the certified values are shown in Fig 1. The results showed that by adding the four different concentrations of ammonia solution, accurate results can be obtained for all the elements tested. However, when the concentration of ammonia solution is less than 0.5%, the resulting test solution is still weakly acidic. Because of the serious damage that $F^-$ can cause to the ICP OES matrix tube in an acidic environment, it is recommended that the ammonia concentration be greater than 0.5%. Considering that excessive alkalinity can reduce the sensitivity of the elements tested [31], an ammonia solution concentration of 0.5% was chosen for extraction in this study.

### Composition of the digestion acid

Nitric acid and hydrofluoric acid are commonly used for soil digestion, and sometimes perchloric acid and sulphuric acid are also used. However, sulphuric acid has a high boiling point and is difficult to remove, which seriously affects the detection efficiency. Therefore, only the combination of nitric acid, hydrofluoric acid and perchloric acid was tested in this study. Perchloric acid is mainly used to remove organic matter during digestion, and the amount added should not be too much, controlled at 1 mL. When dissolving geochemical samples such as

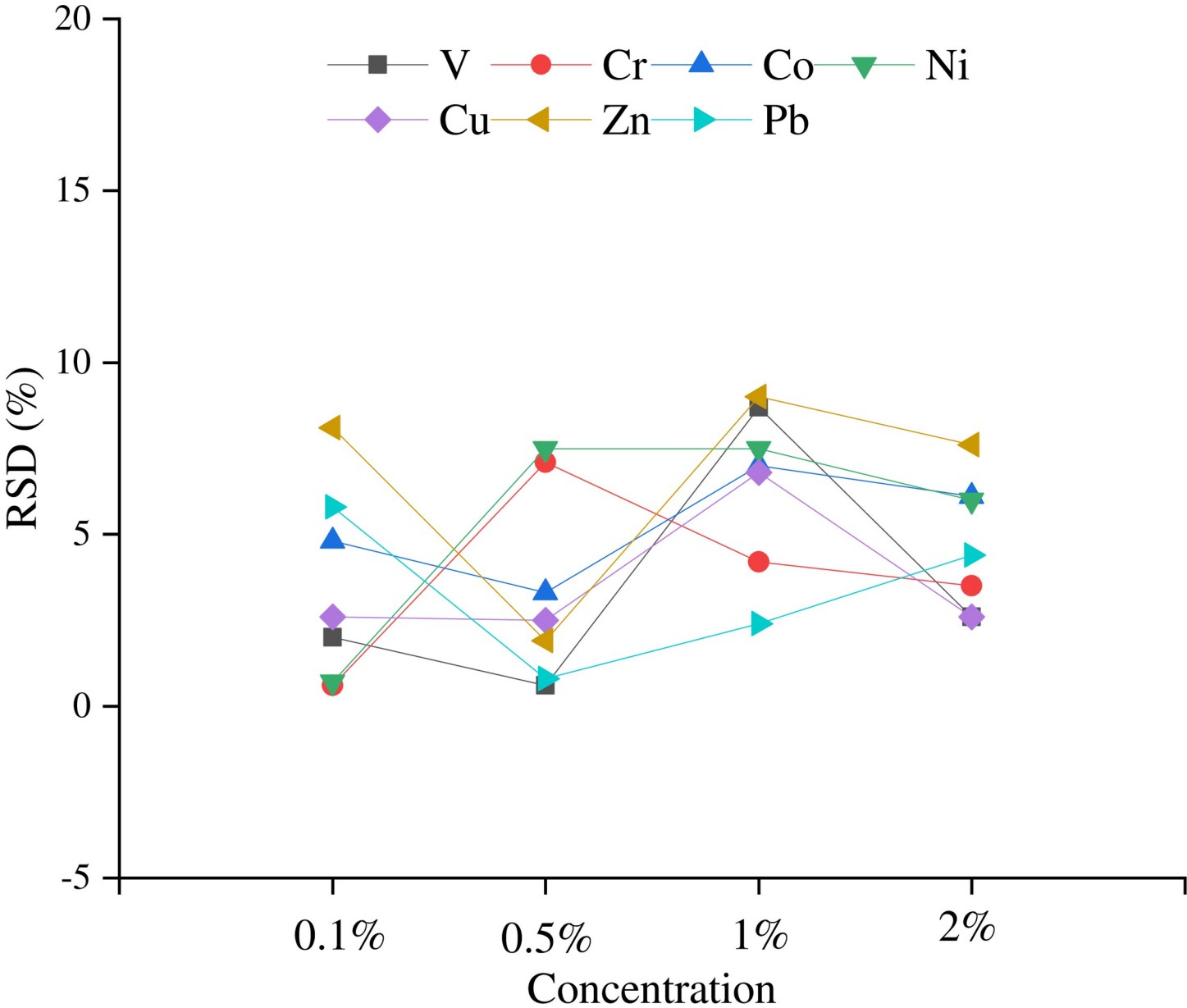

**Fig 1. Effect of ammonia solution concentration on the measured element.**

sediment and soil, the ratio of nitric acid to hydrofluoric acid is usually between 1:1 and 2:3 [32], and combined with the amount of acid added during soil digestion in reference [33], the digestion effect of the following mixed acids was tested in this study. Combination 1: 3 mL nitric acid + 3 mL hydrofluoric acid + 1 mL perchloric acid; Combination 2: 3 mL nitric acid + 5 mL hydrofluoric acid + 1 mL perchloric acid; Combination 3: 5 mL nitric acid + 5 mL hydrofluoric acid + 1 mL perchloric acid; Combination 4: 3 mL nitric acid + 5 mL hydrofluoric acid.

Under the same conditions, the four acid combinations were added to national standard substance GSS1a for digestion and measurement. The results showed that except for combination 1, the other mixed acids gave accurate results. To save acid consumption, a mixed acid of 3 mL nitric acid and 5 mL hydrofluoric acid was chosen for this study.

## Analysis of spectral lines

The selection of ICP OES spectral lines should give full consideration to the detection limit of elements, co-existing element interferences, background interferences and the linear range of the element. For major elements, the linear range and co-existing element interferences are the main factors considered, while for trace elements, the detection limit, co-existing element interferences and background interferences are the main factors considered. After repeated verification, the analytical spectral lines of each element determined in this article are V (292.4), Cr (283.5), Co (228.6), Ni (231.6), Cu (324.7), Zn (213.8) and Pb (220.3). In addition, to minimise the effect of instrument signal drift, standard curves can be redrawn for every 20 samples tested.

## Method quantification limit and precision

Two-point calibration curves were plotted according to the experimental method and the blank solution was measured continuously for 12 times. The detection limits were calculated from the three times the standard deviation of the results and the quantification limits were four times the detection limits. The national standard substance GSS1a was weighed 12 times in parallel and the relative standard deviation (RSD) was calculated according to the experimental method. The method detection limits and precision for V, Cr, Co, Ni, Cu, Zn and Pb were 0.909, 4.32, 0.269, 0.261, 0.968, 3.69 and 2.64 $\mu g\ g^{-1}$, respectively. The precision values were 3.5%, 5.2%, 4.8%, 2.4%, 6.1% and 4.5% for V, Cr, Co, Ni, Cu and Zn, respectively.

## Accuracy of the method

The national standard substances GSS1a, GSS2a, GSS3a, GSS4a, GSS5a and GSS8a were processed according to the experimental method and the results were compared with the certified values. In addition, the "Methods for chemical analysis of silicate rocks—Part 30: Determination of 44 elements" (GB/T 14506.30–2010) was also used to process and determine these national standard substances and the results are shown in Table 2. The results showed that the

**Table 2. Determination results of national standard references of soils.**

|  | GSS1a | | | GSS2a | | | GSS3a | | |
|---|---|---|---|---|---|---|---|---|---|
|  | Certified value/ $\mu g \cdot g^{-1}$ | Measured value/ $\mu g \cdot g^{-1}$ | Measured value*/ $\mu g \cdot g^{-1}$ | Certified value/ $\mu g \cdot g^{-1}$ | Measured value/ $\mu g \cdot g^{-1}$ | Measured value*/ $\mu g \cdot g^{-1}$ | Certified value/ $\mu g \cdot g^{-1}$ | Measured value/ $\mu g \cdot g^{-1}$ | Measured value*/ $\mu g \cdot g^{-1}$ |
| V | 61±4 | 60.7 | 61.0 | 65±5 | 62.0 | 68.0 | 45±3 | 45.1 | 46.9 |
| Cr | 44±3 | 42.2 | 42.7 | 52±4 | 51.6 | 51.7 | 35±3 | 35.5 | 35.1 |
| Co | 10.3±0.6 | 9.98 | 10.3 | 11.1±0.5 | 11.5 | 10.8 | 6.9±0.6 | 6.65 | 6.94 |
| Ni | 16.9±1.5 | 16.1 | 16.6 | 24±2 | 24.5 | 23.8 | 15±1 | 14.2 | 14.6 |
| Cu | 42±5 | 42.2 | 41.5 | 20±2 | 19.9 | 19.1 | 13.4±1.1 | 13.9 | 13.5 |
| Zn | 475±30 | 451 | 489 | 58±3 | 58.7 | 56.2 | 39±3 | 40.0 | 38.2 |
| Pb | 339±12 | 331 | 331 | 27±2 | 25.6 | 26.0 | 28±2 | 27.3 | 28.6 |
|  | GSS4a | | | GSS5a | | | GSS8a | | |
|  | Certified value/ $\mu g \cdot g^{-1}$ | Measured value/ $\mu g \cdot g^{-1}$ | Measured value*/ $\mu g \cdot g^{-1}$ | Certified value/ $\mu g \cdot g^{-1}$ | Measured value/ $\mu g \cdot g^{-1}$ | Measured value*/ $\mu g \cdot g^{-1}$ | Certified value/ $\mu g \cdot g^{-1}$ | Measured value/ $\mu g \cdot g^{-1}$ | Measured value*/ $\mu g \cdot g^{-1}$ |
| V | 125±6 | 128.9 | 127.8 | 136±7 | 140 | 140 | 80±3 | 79.2 | 77.5 |
| Cr | 81±4 | 81.7 | 82.4 | 113±7 | 114 | 111 | 65±4 | 64.5 | 61.8 |
| Co | 20±1 | 19.3 | 19.1 | 18±2 | 18.8 | 18.4 | 12.3±1.0 | 12.0 | 12.6 |
| Ni | 36±2 | 36.2 | 35.7 | 38±2 | 37.7 | 37.4 | 30±2 | 29.0 | 28.5 |

*(Continued)*

**Table 2.** (Continued)

| Cu | 43±2 | 40.8 | 43.9 | 147±10 | 142 | 148 | 24±2 | 23.3 | 23.3 |
| Zn | 92±3 | 93.0 | 91.0 | 172±7 | 177 | 178 | 66±3 | 68.1 | 63.5 |
| Pb | 37±3 | 36.5 | 37.5 | 245±14 | 236 | 241 | 21±2 | 20.1 | 20.0 |

Note: The measured values marked with
"*" are taken using the national recommended method.

measured values of each element in this method are in general agreement with the certified values of the national standard substances and the values measured by the recommended method in China, indicating that the method is fully feasible for the determination of V, Cr, Co, Ni, Cu, Zn and Pb in soil.

## Conclusion

The use of ammonia solution for direct transfer and dilution after timely digestion eliminates the step of stripping hydrofluoric acid, thus improving efficiency. (2) The use of standard materials and blanks to construct two-point calibration curves is not only convenient but also effectively reduces matrix interferences. (3) Hydrofluoric acid and nitric acid are sufficient for complete digestion of the seven trace metals to be tested. (4) The advantage of open digestion is that it is easy to use and efficient. Improvements will further enhance this advantage, especially for the analysis of large batches of samples. (5) The limitations of this method are Some metal elements may precipitate under alkaline conditions, making them difficult to measure. This may result in an incomplete or inaccurate analysis. In addition, the detection limit may be higher under alkaline conditions than under acidic conditions. This means that it may be more difficult to detect low concentrations of metal elements in alkaline solutions.

## Author Contributions

**Data curation:** Jiahan Wang, Junqiao Long.

**Investigation:** Feng Yang.

**Methodology:** Xiujin Yang.

**Project administration:** Wenguang Jiao.

**Supervision:** Cheng Huang.

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
