## [Decision Letter · Decision Letter 0]

26 Jul 2023

PONE-D-23-18937

Open acid dissolution - Ammonia solution extraction - ICP-OES rapid determination of 7 heavy metal elements in soil

PLOS ONE

Dear Dr. Wang,

Thank you for submitting your manuscript to PLOS ONE. After careful consideration, we feel that it has merit but does not fully meet PLOS ONE’s publication criteria as it currently stands. Therefore, we invite you to submit a revised version of the manuscript that addresses the points raised during the review process.

We look forward to receiving your revised manuscript.

Kind regards,

Amitava Mukherjee, ME, Ph.D.

Academic Editor

PLOS ONE

Journal Requirements:

   "This study was financially supported by Geological Survey Project of China Geological Survey (Project No.: DD20220992)."

Reviewers' comments:

Reviewer's Responses to Questions

**Comments to the Author**

1. Is the manuscript technically sound, and do the data support the conclusions?

Reviewer #1: Partly

2. Has the statistical analysis been performed appropriately and rigorously? 

Reviewer #1: No

3. Have the authors made all data underlying the findings in their manuscript fully available?

Reviewer #1: No

4. Is the manuscript presented in an intelligible fashion and written in standard English?

Reviewer #1: No

5. Review Comments to the Author

Reviewer #1: Dear authors/editor,

The manuscript entitled “Open acid dissolution - Ammonia solution extraction - ICP-OES rapid determination of 7 heavy metal elements in soil” report on a new method for to overcome the corrosion of hydrofluoric acid on the ICP-OES injection system in the acid dissolution system. It presents scientific relevance for the area of Chemistry, Biology, and others area.

The manuscript presents an interesting proposal, but from determining V, Cr, Co, Ni, Cu, Zn and Pb if digestion with HF is necessary? These analytes can be extracted with a mixture of strong acids (such as HNO3, HCl, H2SO4, etc…) and oxidizing agents! If the proposal were to determine silicon, the use of HF is justified!

The language (English) are satisfactory (I suggest the final revision)! However, you need to change some details/information in the abstract, Introduction, Methods, results, discussion and conclusions.

1. Title: Adequate! To replace “ICP-OES” by “ICP OES”, and throughout the entire manuscript.

2. Abstract: Adequate, but I suggest rewrite and add information:

- To replace “μg/g” by “mL min-1” and “μg g-1”, and throughout the entire manuscript.

- What are the quantification limit values?

- I suggest highlighting the "innovative" proposal of the study.

3. Introduction section: It is well written, but I suggest:

- The term “heavy metals” is not appropriate. The term “heavy metals” is the subject of many discussions. I suggest that the term be replaced, throughout the manuscript, according to [Science of the Total Environment 610–611 (2018) 419–420: “Heavy metal” - What to do now: To use or not to use?] and [Hazrat Ali & Ezzat Khan (2018) What are heavy metals? Long-standing controversy over the scientific use of the term ‘heavy metals’ – proposal of a comprehensive definition, Toxicological & Environmental Chemistry, 100:1, 6-19, DOI: 10.1080/02772248.2017.1413652]. Therefore, I strongly suggest removal of “heavy metals” from all text and replacement in the abstract and full text of submitted paper with words like “potentially toxic metal(s)/element(s)” or “trace metal(s)/element(s)”, according to the context, throughout the manuscript.

- I suggest inserting more information and references about: To avoid damage to the ICP OES injection system from hydrofluoric acid; and, alkaline fusion methods.

- I suggest highlighting the "innovative" proposal of the study, as well as the advantages / disadvantages, at the end of the introduction.

4. Material and Methods section:

- I suggest inserting more information about the mineral composition of the national standard substances.

- In “Experimental method”: I request more information about the sample storage time until analysis, etc. How were the digestion conditions optimized? Did you follow any published article/pervious protocol? If yes, insert reference!

- In “Experimental method”: The authors wrote: “The national standard substance GSS2, digested simultaneously with the sample, was used as the peak and the blank solution as the valley. The two-point standard working curve for each element was obtained by measuring the solutions under the selected instrument operating conditions. The test solution was then measured under the same conditions”. I suggest a better explanation for: "peak and the blank solution as the valley" and "The two-point standard working curve for each element...". Was the curve plotted with only 2 points? How were the precision tests carried out? How many concentration levels were used? The authors proposed the validation of an analytical method, robustness, etc. What concentration levels are used to assess accuracy? Ideally, 3 concentration levels (low, medium, and high) should be used to assess accuracy. Robustness? I suggest detailing the proposed method in more detail...

- The authors proposed the validation of an analytical method, however there are no details about this step! What are the analytical validation parameters used? Has the proposed method been validated? If so, which protocol / guidelines (IUPAC, ICH, ETC.) did you follow? What are the validation parameters studied? Precision, accuracy, LOD, LOQ, robustness, etc. What concentration levels are used to assess accuracy? I suggest detailing the proposed method in more detail...

5. Results and discussion section:

- In “Calibration curves” section: The authors wrote: “Therefore, in this study, the national standard substance GSS2 was used as the peak and the process blank as the valley to draw the calibration curves.” I suggest expanding the discussions with more details!

- In “Concentration of ammonia solution” section: How were the ammonia concentrations selected? Did you follow any published article/pervious protocol? If yes, insert reference!

- In “Composition of the digestion acid” section: How were acid combinations defined? Univariate analysis? Multivariate? Did you follow any published article/pervious protocol? If yes, insert reference!

- In “Method detection limit and precision” section: And the quantification limits?

I suggest expanding the discussions, comparing the data with the scientific literature. I suggest, at the end of the section, to write a paragraph summarizing the findings and their impacts on the research proposal.

6. Material and Methods section:

- Conclusion section: The conclusion is written in topics!!!??? I suggest highlighting the "innovative" proposal of the study, as well as the advantages/ disadvantages/limitations.

* Tables and Figures: Few tables and fihures.

* References: Please, check if the references are in accordance with the journal's rules.

6. PLOS authors have the option to publish the peer review history of their article (what does this mean?). If published, this will include your full peer review and any attached files.

Reviewer #1: No

---

## [Decision Letter · Decision Letter 1]

4 Sep 2023

PONE-D-23-18937R1Open acid dissolution - Ammonia solution extraction - ICP-OES rapid determination of 7 heavy metal elements in soilPLOS ONE

Dear Dr. Wang,

Thank you for submitting your manuscript to PLOS ONE. After careful consideration, we feel that it has merit but does not fully meet PLOS ONE’s publication criteria as it currently stands. Therefore, we invite you to submit a revised version of the manuscript that addresses the points raised during the review process.

We look forward to receiving your revised manuscript.

Kind regards,

Amitava Mukherjee, ME, Ph.D.

Academic Editor

PLOS ONE

Journal Requirements:

Reviewers' comments:

Reviewer's Responses to Questions

**Comments to the Author**

1. If the authors have adequately addressed your comments raised in a previous round of review and you feel that this manuscript is now acceptable for publication, you may indicate that here to bypass the “Comments to the Author” section, enter your conflict of interest statement in the “Confidential to Editor” section, and submit your "Accept" recommendation.

Reviewer #1: All comments have been addressed

2. Is the manuscript technically sound, and do the data support the conclusions?

Reviewer #1: Partly

3. Has the statistical analysis been performed appropriately and rigorously? 

Reviewer #1: No

4. Have the authors made all data underlying the findings in their manuscript fully available?

Reviewer #1: Yes

5. Is the manuscript presented in an intelligible fashion and written in standard English?

Reviewer #1: Yes

6. Review Comments to the Author

Reviewer #1: The authors improved the manuscript based on the suggestions.

I suggest expanding the "discussions", comparing the results with the scientific literature. In the discussion only 3 references are cited (30-32)! In "Response to Reviewers" authors wrote "The use of a mixed acid solution containing hydrofluoric acid (HF) for dissolution and subsequent measurement by ICP-OES or ICP-MS is the primary method for determining the values of almost all soil standard substances in China. If other methods, such as those used by the

EPA in other countries, which involves extraction with nitric acid (HNO3) and hydrochloric acid (HCl) followed by measurement, are used, the measured values may not be consistent with the certified values of Chinese standard substances. Therefore, hydrofluoric acid was used as the dissolving agent in this study". Therefore, it is necessary to compare the manuscript data with these studies.

Insert the values of standard deviations (+/- sd) in Table 2!

7. PLOS authors have the option to publish the peer review history of their article (what does this mean?). If published, this will include your full peer review and any attached files.

Reviewer #1: No

---

## [Editor Report · Decision Letter 2]

14 Sep 2023

Open acid dissolution - Ammonia solution extraction - ICP OES rapid determination of 7 trace metal elements in soil

PONE-D-23-18937R2

Dear Dr. Wang,

We’re pleased to inform you that your manuscript has been judged scientifically suitable for publication and will be formally accepted for publication once it meets all outstanding technical requirements.

Kind regards,

Amitava Mukherjee, ME, Ph.D.

Academic Editor

PLOS ONE
---

## [Editor Report · Acceptance letter]

25 Sep 2023

PONE-D-23-18937R2 

Open acid dissolution - Ammonia solution extraction - ICP OES rapid determination of 7 trace metal elements in soil 

Dear Dr. Wang:

I'm pleased to inform you that your manuscript has been deemed suitable for publication in PLOS ONE. Congratulations! Your manuscript is now with our production department. 

Kind regards, 

on behalf of

Professor Dr. Amitava Mukherjee 

Academic Editor

PLOS ONE